# CURRICULUM-INSPIRED TRAINING FOR SELECTIVE NEURAL NETWORKS

## ABSTRACT

We consider the problem of training neural network models for selective classification, where the models have the reject option to abstain from predicting certain examples as needed. Recent advances in curriculum learning have demonstrated the benefit of leveraging the example difficulty scores in training deep neural networks for typical classification settings. Example difficulty scores are even more important in selective classification as a lower prediction error rate can be achieved by rejecting hard examples and accepting easy ones. In this paper, we propose a curriculum-inspired method to train selective neural network models by leveraging example difficulty scores. Our method tailors the curriculum idea to selective neural network training by calibrating the ratio of easy and hard examples in each mini-batch, and exploiting difficulty ordering at the mini-batch level. Our experimental results demonstrate that our method outperforms both the state-of-the-art and alternative methods using vanilla curriculum techniques for training selective neural network models.

## 1 INTRODUCTION

In *selective classification*, the goal is to design a predictive model that is allowed to abstain from making a prediction whenever it is not sufficiently confident. A model with this reject option is called a *selective model*. In other words, a selective model will reject certain examples as appropriate, and provide predictions only for accepted examples. In many real-life scenarios[1], such as medical diagnosis, robotics and self-driving cars (Kompa et al., 2021), selective models are used to minimize the risk of wrong predictions on the hard examples by abstaining from providing any predictions and possibly seeking human intervention.

In this paper, we focus on selective neural network models, which are essentially neural network models with the reject option. These models have been shown to achieve impressive results (Geifman & El-Yaniv, 2019; 2017; Liu et al., 2019). Specifically, Geifman & El-Yaniv (2019) proposed a neural network model, SELECTIVENET, that allows end-to-end optimization of selective models. SELECTIVENET contains a main body block followed by three heads: one for minimizing the error rate among the accepted examples, one for selecting the examples for acceptance or rejection, and one for the auxiliary task of minimizing the error rate on all examples. These three heads are illustrated later in Figure 1. The final goal of this model is to minimize the error rate among the accepted examples while satisfying a coverage constraint in terms of the least percentage of examples that need to be accepted. The coverage constraint is imposed to avoid the trivial solution of rejecting all examples to get a $0\%$ error rate. Ideally, the model should reject hard examples and accept easy ones to lower its overall error rate. While it is clear that difficulty scores are helpful, they are typically unknown in most settings. Therefore, to leverage difficulty scores we must overcome two challenges: (1) how to obtain the difficulty scores as accurately as possible and (2) how to best utilize them in a selective neural network model.

Recent curriculum learning techniques have investigated how to use example difficulty scores to improve neural network models' performance (Hacohen & Weinshall, 2019; Wu et al., 2020). To the best of our knowledge, these techniques only consider the typical classification setting where the error rate on all examples should be minimized. Curriculum learning techniques often use a

---

[1]These are further elaborated in Section A.1 in the appendix.

scoring function to estimate the difficulty scores. There are different approaches to constructing a scoring function from a reference model, such as (1) confidence score (Hacohen & Weinshall, 2019), (2) learned epoch/iteration (Wu et al., 2020), and (3) estimated c-score (Jiang et al., 2020). Prior work has shown that these scoring functions are highly correlated with one another and lead to similar performance (Wu et al., 2020). Existing curriculum learning techniques use the estimated difficulty scores only to decide the order of examples being exposed to the model during the training process. Typically, they expose easy examples to the model during the early phase of the training process, and gradually transition to the hard examples as the training progresses. In this paper, we consider selective classification, which has a more relaxed goal: minimize the error rate on *the accepted examples* for a given coverage constraint. Difficulty scores are even more useful in selective classification, because the selection of accepted examples has a direct impact on the model's final performance (i.e., a lower error rate can be achieved by rejecting hard examples and accepting easy ones). This inspires us to design a new method for training selective neural network models.

**Contributions.** In this paper, we propose a new method for training selective neural network models inspired by curriculum learning. Our curriculum-inspired method has two benefits. First, existing training methods ignore the coverage constraint when constructing the mini-batch for each iteration, which we show will introduce misguiding noise to the training progress of selective neural network models. Taking advantage of the estimated difficulty scores, our method calibrates the ratio of easy and hard examples in each mini-batch to match the desirable value due to the coverage constraint. Second, our method also adopts the curriculum idea of increasing difficulty levels gradually over the training process, which has been demonstrated to improve neural network training (Hacohen & Weinshall, 2019; Wang et al., 2021). However, instead of relying on existing vanilla curriculum techniques that exploit difficulty ordering at the example level, our method is tailored to selective neural network training by exploiting difficulty ordering at the mini-batch level. We will show that this change is necessary because of the need to match coverage constraint as just explained.

To summarize, we make the following contributions:

1. To the best of our knowledge, we are the first to investigate the benefits of leveraging example difficulty scores to improve selective neural network model training.

2. We design a new curriculum-inspired method with two benefits for training selective neural networks: (a) calibrating the ratio of easy and hard examples in each mini-batch, which has been ignored by existing methods (Section 4.1); (b) adopting a curriculum idea which is tailored to selective neural network training by exploiting difficulty ordering at the mini-batch level (Section 4.2).

3. We conduct extensive experiments demonstrating that our method improves the converged error rate (up to $13\%$ lower) of selective neural network models compared to the state-of-the-art (Section 5.2). We also show that our method is better than alternative designs using vanilla techniques from existing curriculum learning literature (Section 5.3).

## 2 RELATED WORK

**Selective classification.** Prior work on selective classification primarily focuses on adding the reject option to classical learning algorithms such as SVM (Fumera & Roli, 2002), nearest neighbors (Hellman, 1970), boosting (Cortes et al., 2016) and online learning methods (Cortes et al., 2018). In particular, (Geifman & El-Yaniv, 2017) applies selective classification techniques in the context of deep neural networks (DNNs). They show how to construct a selective classifier given a trained neural network model. They decide whether or not to reject each example based on a confidence score. They rely on two techniques for extracting confidence scores from a neural network model: Softmax Response (SR), and Monte-Carlo dropout (MC-dropout). SR is the maximal activation in the softmax layer for a classification model, which is used as the confidence score. MC-dropout estimates the confidence score based on the statistics of numerous forward passes through the network with the dropout applied. Unfortunately, MC-dropout requires hundreds of forward passes for each example, incurring a massive computational overhead. More recently, Geifman & El-Yaniv (2019) proposes a selective neural network that jointly learns a predictive function and a selection function. This model is trained end-to-end, resulting in a selective model that is optimized over the covered domain. They show empirically that this selective neural network outperforms previous methods based on SR or MC-dropout. In addition, inspired by portfolio theory, Liu et al. (2019) propose a

new loss function for selective classification based on the doubling rate of gambling. Self-adaptive training (Huang et al., 2020) that calibrates training process by model predictions can also be applied to selective classification. In Gangrade et al. (2021), the selective classification is formulated into optimising a collection of class-wise decoupled one-sided empirical risks. However, none of the prior work in selective classification has exploited estimated difficulty scores as in curriculum learning, which is what we propose for the first time in this paper.

**Curriculum learning.** Inspired by how human students are gradually introduced to concepts with increasing complexity, curriculum learning techniques train neural networks by feeding easier examples earlier in the training process and gradually transitioning to more difficult examples. There is a huge surge of interests in applying curriculum learning to different domains in recent years, e.g., image classification (Guo et al., 2018), machine translation (Platanios et al., 2019), language model pre-training (Li et al., 2021), healthcare prediction (El-Bouri et al., 2020) and graph learning (Gong et al., 2019). Despite the differences among domains, these techniques follow a general pattern where a sequence of training criteria are presented to the model over $T$ training steps and the training criteria affect how examples are selected at each step (Wang et al., 2021; Soviany et al., 2022). Typically, example selection works in the following way: the mini-batch at each iteration is uniformly sampled from a working subset, which starts with a few easiest examples, and is being appended with harder and harder examples after every several epochs until all the training examples are added. Our curriculum-inspired method is also based on the above example selection strategy. Note that *anti-curriculum* method uses the opposite strategy by starting from the hardest examples and transitioning to easier examples. We do not consider the anti-curriculum strategy because it has been shown to have worse performance than normal curriculum strategy in recent studies (Wu et al., 2020; Hacohen & Weinshall, 2019). Similarly, another type of strategies for example selection that is different from curriculum learning is example weighting (Shrivastava et al., 2016; Katharopoulos & Fleuret, 2018). However, some studies (Wang et al., 2021; Soviany et al., 2022) have shown that example weighting methods are more prone to noise and outliers in the datasets, because they tend to select the hardest examples which might actually be noise and outliers. In this paper, we focus on curriculum learning due to its suitability for handling complex real-world scenarios where selective classification is usually applied.

## 3 BACKGROUND

In this section we describe the problem formulation for selective classification, the selective neural network model and a baseline method for training it.

### 3.1 SELECTIVE CLASSIFICATION PROBLEM FORMULATION

Here, we consider the multi-class classification setting. Let $\mathcal{X}$ be the input feature space, $\mathcal{Y}$ the label space and $P(\mathcal{X}, \mathcal{Y})$ a distribution over $\mathcal{X} \times \mathcal{Y}$. A predictive model is a function $f : \mathcal{X} \to \mathcal{Y}$. Given a loss function $l : \mathcal{Y} \times \mathcal{Y} \to \mathbb{R}^+$, the risk of the model $f$ is $\mathbb{E}_{P(\mathcal{X}, \mathcal{Y})}[l(f(x), y)]$. In the classical supervised learning, the goal is to find the model $f$ such that the model risk is minimized.

In this paper, we consider the selective model which is a pair of functions $(f, g)$ where $f : \mathcal{X} \to \mathcal{Y}$ is the predictive model, and $g : \mathcal{X} \to \{0, 1\}$ is a selection function. The selection function is used to decide whether each example $x$ is accepted or not:

$$(f, g)(x) = \begin{cases} f(x), & \text{if } g(x) = 1; \\ \text{ABSTAIN}, & \text{if } g(x) = 0. \end{cases} \tag{1}$$

Thus, the selective model rejects (i.e., abstains from) an example $x$ iff $g(x) = 0$. Note that a soft selection function $g : \mathcal{X} \to [0, 1]$ can also be considered, from which decisions are taken probabilistically or deterministically using a threshold. The selective risk of $(f, g)$ is

$$R(f, g) = \frac{\mathbb{E}_P[l(f(x), y)g(x)]}{\mathbb{E}_P[g(x)]}. \tag{2}$$

The coverage $\phi(g)$ is defined as $\mathbb{E}_P[g(x)]$, i.e., the ratio of the number of accepted examples among all examples. Clearly, there is a trade-off between risk and coverage. In other words, a lower risk could be achieved by sacrificing the coverage. The entire performance of a selective model can be

measured via its risk-coverage curve (El-Yaniv et al., 2010). As with Geifman & El-Yaniv (2019), we adopt the goal of finding the model $(f, g)$ such as its selective risk is minimized given a target coverage $c$.

Given a labeled set $S = \{(x_i, y_i)\}_{i=1}^m$, there are empirical counterparts for the selective risk and coverage. The empirical selective risk is

$$\hat{r}(f, g|S) = \frac{\frac{1}{m} \sum_{i=1}^m l(f(x_i), y_i) g(x_i)}{\hat{\phi}(g|S)}, \tag{3}$$

and the empirical coverage is

$$\hat{\phi}(g|S) = \frac{1}{m} \sum_{i=1}^m g(x_i). \tag{4}$$

## 3.2 SELECTIVE NEURAL NETWORK MODEL

In this paper, we focus on the neural network model SELECTIVENET from Geifman & El-Yaniv (2019) because of its impressive empirical performance. SELECTIVENET optimizes both $f$ and $g$ in a single neural network model. The main body block can be any type of architecture that is typically used for solving the problem at hand. For example, convolutional neural networks or residual neural networks are typically used for image classification (Krizhevsky et al., 2012; He et al., 2016), and transformer networks are typically used for natural language understanding (Devlin et al., 2018). SELECTIVENET has three output heads for prediction $f$, selection $g$ and auxiliary prediction $h$, respectively. The prediction head $f$ and selection head $g$ are the two functions previously defined defined in Section 3.1. The auxiliary head $h$ handles a prediction task with the goal of exposing the main body block to all training examples throughout the training process. As with Geifman & El-Yaniv (2019), the auxiliary head $h$ uses the same prediction task assigned to $f$ in this paper. We only need $h$ during training by adding its loss function (defined later in Equation 6) to the main selective classification loss (Equation 5). Given a training set $S = \{x_i, y_i\}_{i=1}^m$ and a target coverage $c$, the selective training objective for the heads $f$ and $g$ is

$$\mathcal{L}_{(f,g)} = \hat{r}(f, g) + \lambda \max(0, c - \hat{\phi}(g|S))^2 \tag{5}$$

where $\lambda$ is a hyper-parameter. The standard loss function is used for auxiliary head $h$:

$$\mathcal{L}_h = \frac{1}{m} \sum_{i=1}^m l(h(x_i), y_i). \tag{6}$$

Thus, the overall training objective is

$$\mathcal{L} = \alpha \mathcal{L}_{(f,g)} + (1 - \alpha) \mathcal{L}_h, \tag{7}$$

where $\alpha$ is a hyper-parameter controlling the relative importance of the above two losses. A stochastic gradient descent (SGD)-type optimization method is used to minimize the above objective (Geifman & El-Yaniv, 2019), which is an iterative process. At each iteration, a mini-batch of examples are picked from the entire training set, which are used to compute the gradient to update the model parameters. We treat this training method as our baseline, and propose a new method inspired by curriculum learning in the next section.

It is worth noting that, **regardless of the value of the target coverage $c$, all examples participate and are accounted for in minimizing our training objective (Equation 7)**. This is because the selection head $g$ (used to decide whether an example should be accepted or not) is trained over *all* examples. Furthermore, the auxiliary head is optimized over *all* examples as it has shown to improve the main body block training (Geifman & El-Yaniv, 2019). Once the training is completed, the auxiliary head is no longer needed. We keep the prediction head $f$ and selection head $g$ together with the main body block, and use them for selective classification according to Equation 1.

## 4 CURRICULUM-INSPIRED TRAINING METHOD

Inspired by curriculum learning, we propose a new method for training the selective neural network model. Our method is presented in Algorithm 1. The key idea here is to (1) construct mini-batches

by calibrating the ratio of easy and hard examples, and (2) exploit the difficulty ordering at the mini-batch level, i.e., control the order in which the mini-batches are exposed by gradually narrowing the difficulty gap between easy and hard examples within each mini-batch. These steps are discussed in Sections 4.1 and 4.2, respectively. In Section 4.3, we discuss our approach to estimate the example difficulty scores.

## 4.1 MINI-BATCH CONSTRUCTION

Since we are using a stochastic gradient descent (SGD)-type optimization method to train the selective neural network model, we need to construct a mini-batch of examples to be exposed at each iteration. We can show[2] that, when the mini-batch has the proportion of easy examples different from the target coverage, as a result of minimizing the loss during the training, even an already perfect selection function tends to move to the direction where mistakes will be made (i.e., reject easy examples or accept hard examples). Therefore, a mini-batch which has the proportion of easy examples different from the target coverage will misguide the selection function regarding which examples should be accepted or rejected, which further affects the prediction head's learning

---

**Algorithm 1** Curriculum-inspired method for training selective neural network.

1: **Input**: training set $S$, target coverage $c$, mini-batch size $b$, pacing function $p$ and scoring function $q$
2: **Output**: sequence of mini-batches $[B_1, B_2, \cdots]$
3: Sort $S$ according to $q$ in ascending order   //easier examples are before the harder examples
4: $result \leftarrow []$
5: **for** $i \leftarrow 1$ **to** $T$ **do**   //$T$ is the total number of iterations
6:     $size \leftarrow p(i)$;
7:     $size_E \leftarrow \lceil p(i) * c \rceil$;   //the size of easy active set $S_E$
8:     $size_H \leftarrow size - size_E$;   //the size of hard active set $S_H$
9:     $S_E = S[: size_E]$;   //select the first $size_E$ examples from the ordered training set $S$
10:    $S_H = S[-size_H :]$;   //select the last $size_H$ examples from ordered training set $S$)
11:    $B_E \leftarrow$ uniformly sample $\lceil b * c \rceil$ examples from $S_E$;
12:    $B_H \leftarrow$ uniformly sample $b - \lceil b * c \rceil$ examples from $S_H$;
13:    $B_i = B_E \bigcup B_H$;   //form the mini-batch for the $i$-th iteration
14:    append $B_i$ to $result$;
15: **end for**
16: **Return:** $result$

---

progress. Fortunately, we can leverage the estimated difficulty scores to alleviate this issue. Specifically, we partition all the training examples into an easy set and a hard set according to their estimated scores. Assume the mini-batch size is $b$, the total number of iterations is $T$ and the total number of training examples is $m$. The easy set contains $m * c$ examples with the lowest difficulty scores, whereas the hard set contains the rest of the examples. We construct the mini-batch by selecting $\lceil b * c \rceil$ examples from the easy set, called $B_E$, and $b - \lceil b * c \rceil$ examples from the hard set, called $B_H$, as shown in Lines 11 and 12 of Algorithm 1. The union of $B_E$ and $B_H$ becomes the mini-batch $B_i$ for the $i$-th iteration (Algorithm 1 Line 13). Here, we use the $\lceil \cdot \rceil$ operator because $b * c$ may not be an integer. By calibrating the ratio of easy and hard examples in each mini-batch in the above manner, we can ensure that the proportion of easy examples is roughly $c$.

## 4.2 MINI-BATCH ORDERING

Curriculum learning techniques suggest exposing easy examples to the model early on in the training process, and gradually transitioning to hard examples. We still adopt the curriculum idea, but we argue that in a selective neural network context, curriculum should be designed at the batch level. We cannot expose examples exactly the same way that curriculum learning does because our mini-batch should include both easy and hard examples with their ratio appropriately calibrated to meet the coverage constraint, as explained earlier in subsection 4.1. We therefore need to generalize curriculum learning to impose the ordering at the mini-batch level, i.e., by exposing easy mini-batches to the model early in the training process. The key intuition is as follow. **We consider a mini-batch easy for the model (especially its selection head) if its constituent easy examples are clearly separate from its constituent hard examples in terms of their difficulty scores.** Similarly, we consider a mini-batch hard if its constituent easy examples are close to its constituent hard examples in terms of

---

[2]Detailed explanations are included in Section A.2 in the appendix.

their difficulty scores. We propose to expose easy mini-batches to the model early in the training and gradually transition to hard mini-batches. This enables us to gradually narrow down the difficulty gap between easy and hard examples. Similar to other curriculum learning techniques (Hacohen & Weinshall, 2019), we also need to provide two functions as the input to our curriculum-inspired method: a pacing function $p$ and a scoring function $q$. A pacing function $p$ is used to control the curriculum pace, i.e., the speed at which we transition from easy to hard mini-batches. A scoring function $q$ is used to indicate how hard each example is (i.e., estimated difficulty score), which will be explained in the next subsection.

Specifically, the pacing function $p$ will output an integer $p(i)$ for each iteration $i$, which we then use to decide how many examples to consider via uniform sampling when constructing the $i$-th mini-batch (Algorithm 1 Line 6). In this paper, we use the fixed exponential pacing function from Hacohen & Weinshall (2019) because of its good performance in practice. At the $i$-th iteration, instead of picking the $p(i)$ easiest examples from the training set, we choose $\lceil p(i) * c \rceil$ easiest examples as the *easy active set*, called $S_E$, and $p(i) - \lceil p(i) * c \rceil$ hardest examples as the *hard active set*, called $S_H$ (Algorithm 1 Line 9 and Line 10). Then, we uniformly sample $\lceil b * c \rceil$ examples from $S_E$, called $B_E$, and uniformly sample $b - \lceil b * c \rceil$ examples from $S_H$, called $B_H$ (Algorithm 1 Line 11 and Line 12). The union of $B_E$ and $B_H$ becomes the mini-batch $B_i$ for the $i$-th iteration (Algorithm 1 Line 13). The entire process is also illustrated in Figure 1.

### 4.3 DIFFICULTY SCORE ESTIMATION

We use the following approach for getting the scoring function $q$, which outputs a scalar value $q(x)$ for each example $x$ as its estimated difficulty score. First, we train the same model using uniformly sampled mini-batches (i.e., vanilla method), and run each example through it to get the activation levels of its main body output as its condensed representation. We then run an outlier detection algorithm, Minimum Covariance Determinant (MCD) (Rousseeuw, 1984), to compute the Mahalanobis distance for each example with respect to all examples that belong to the same class with this example. The idea behind MCD is to find a subset of examples whose empirical covariance has the smallest determinant, hence resulting in the tightest representation of the overall dataset, which may otherwise be quite noisy and have outliers. This subset of examples is then used to compute the center location

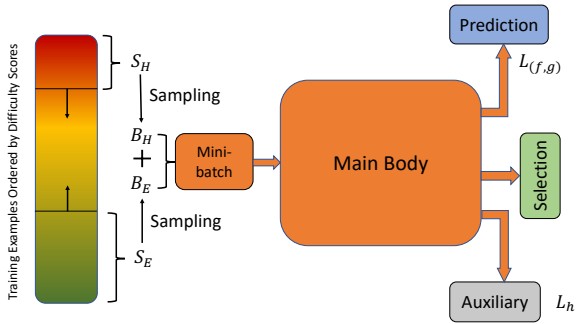

Figure 1: Diagram of our curriculum-inspired method for training the selective neural network model. The training examples are ordered by their difficulty scores as shown in the left-most box. Hard examples are at the top (red color) and easy examples are at the bottom (green color). At each iteration, the mini-batch is constructed by sampling from both the hard part and the easy part, with their ratio appropriately calibrated. As training process goes, we gradually narrow the difficulty gap between easy and hard examples, indicated by the boundaries of the easy and hard parts moving towards each other in the diagram.

$\mu$ and covariance $\Sigma$. For any example $x$, its Mahalanobis distance is computed as $d(x) = (x - \mu)^T \Sigma^{-1} (x - \mu)$. The larger the Mahalanobis distance $d(x)$, the noisier the example $x$. We thus treat $d(x)$ as the scoring function output $q(x)$ for example $x$, i.e., $q(x) = d(x), \forall x$.

In practice, the overhead of this new approach for difficulty score estimation should not become a concern. Firstly, the model using uniform sampling (i.e., without rejection) needs much less iterations to get an estimation of difficulty scores than the model with rejection. In all of our experiments, we use half of the iterations compared to training the model with rejection. Secondly, we run the MCD separately for the examples belonging to each class. The number of examples in each class is typically small enough for running MCD (e.g., 5k for CIFAR10, 7k for SVHN and 500 for CIFARI100). Both factors limit the overhead of our approach. What's more, the overhead of getting difficulty score estimation can be greatly reduced by simply using an exiting approach based on

Table 1: Results of moderate-sized CNN model on CIFAR10. Testing classification error rates (in %) are shown for different target coverage rates. The relative improvement of error rates of our method w.r.t. SelectiveNet is shown at the last column.

| Coverage | Error Rate | | Impr. |
|---|---|---|---|
| | Our Method | SelectiveNet | |
| 0.95 | $\mathbf{7.32 \pm 0.05}$ | $7.57 \pm 0.1$ | 3.30% |
| 0.9 | $\mathbf{5.63 \pm 0.04}$ | $6.07 \pm 0.1$ | 7.25% |
| 0.85 | $\mathbf{4.30 \pm 0.04}$ | $4.53 \pm 0.08$ | 5.08% |
| 0.8 | $\mathbf{3.02 \pm 0.02}$ | $3.47 \pm 0.07$ | 12.97% |
| 0.75 | $\mathbf{2.16 \pm 0.04}$ | $2.45 \pm 0.07$ | 11.84% |
| 0.7 | $\mathbf{1.47 \pm 0.02}$ | $1.67 \pm 0.08$ | 11.98% |

Table 2: Results of moderate-sized CNN model on CIFAR100. Testing classification error rates (in %) are shown for different target coverage rates. The relative improvement is shown at the last column.

| Coverage | Error Rate | | Impr. |
|---|---|---|---|
| | Our Method | SelectiveNet | |
| 0.95 | $\mathbf{30.63 \pm 0.06}$ | $31.13 \pm 0.22$ | 1.61% |
| 0.9 | $\mathbf{28.95 \pm 0.08}$ | $29.05 \pm 0.14$ | 0.35% |
| 0.85 | $\mathbf{27.07 \pm 0.08}$ | $27.32 \pm 0.09$ | 0.92% |
| 0.8 | $\mathbf{25.00 \pm 0.05}$ | $25.39 \pm 0.07$ | 1.54% |
| 0.75 | $\mathbf{23.16 \pm 0.03}$ | $23.67 \pm 0.05$ | 2.15% |
| 0.7 | $\mathbf{21.28 \pm 0.02}$ | $21.97 \pm 0.03$ | 3.14% |

off-the-shelf pre-trained models (e.g., **Inception** in our ablation study in Section 5.3). Equipped with either approach for getting difficulty score estimation, our method can outperform the baseline. However, our new difficulty estimation approach leads to better empirical performance. We think this is because MCD offers the robustness needed for handling complex datasets (which might be noisy with outliers).

## 5 EXPERIMENTS

In this section, we evaluate our proposed method and compare it against a state-of-the-art method for training selective neural network models. Similar to Geifman & El-Yaniv (2019), we also report both the testing classification error rates (which is the same as selective risk values when using 0-1 loss) as well as the relative improvements of the error rates. In summary, our experiments show the following [3]: (1) our method achieves a lower error rate (up to 10% lower) than the state-of-the-art for the same target coverage rate for a variety of different main body network architectures and datasets (section 5.2); (2) our method is better than other alternative designs using difficulty scores or example selection from existing curriculum learning literature (section 5.3).

### 5.1 SETUP

**Datasets.** Similar to prior work, we use the following three datasets: **CIFAR10** (Krizhevsky et al., 2009), **CIFAR100** (Krizhevsky et al., 2009) and **SVHN** (Netzer et al., 2011) in our experiments.

**Baseline method and architectures.** We compare our method against SELECTIVENET (Geifman & El-Yaniv, 2019), which is considered the state-of-the-art. We vary the target coverage rate $c$ with 6 different values: $0.95, 0.9, 0.85, 0.8, 0.75, 0.7$. We use two different neural network architectures for the main body of our selective neural network model: moderate-sized convolutional network architecture (Hacohen & Weinshall, 2019) and VGG-16 architecture (Simonyan & Zisserman, 2014). Details about these architectures can be found in the appendix.

---

[3]In the appendix, we also show that (i) our method has a faster convergence speed (up to 50% faster) than the state-of-the-art (section A.5), and (ii) our method can reduce the coverage violation (i.e., the difference between target coverage and empirical coverage) by more than 10% (section A.6)

Table 3: Results of moderate-sized CNN model on SVHN. Testing classification error rates (in %) are shown for different target coverage rates. The relative improvement is shown at the last column.

| Coverage | Error Rate | | Impr. |
|---|---|---|---|
| | Our Method | SelectiveNet | |
| 0.95 | $\mathbf{1.91 \pm 0.03}$ | $2.08 \pm 0.02$ | 8.17% |
| 0.9 | $\mathbf{1.03 \pm 0.04}$ | $1.12 \pm 0.06$ | 8.04% |
| 0.85 | $\mathbf{0.63 \pm 0.02}$ | $0.66 \pm 0.01$ | 4.55% |
| 0.8 | $\mathbf{0.49 \pm 0.03}$ | $0.51 \pm 0.02$ | 3.92% |
| 0.75 | $\mathbf{0.41 \pm 0.01}$ | $0.42 \pm 0.03$ | 2.38% |
| 0.7 | $\mathbf{0.35 \pm 0.01}$ | $0.39 \pm 0.01$ | 10.26% |

Table 4: Results of VGG model on CIFAR100. Testing classification error rates (in %) are shown for different target coverage rates. The relative improvement is shown at the last column.

| Coverage | Error Rate | | Impr. |
|---|---|---|---|
| | Our Method | SelectiveNet | |
| 0.95 | $\mathbf{28.41 \pm 0.07}$ | $28.92 \pm 0.09$ | 1.76% |
| 0.9 | $\mathbf{25.67 \pm 0.04}$ | $25.78 \pm 0.02$ | 0.43% |
| 0.85 | $\mathbf{22.94 \pm 0.02}$ | $23.56 \pm 0.08$ | 2.63% |
| 0.8 | $\mathbf{21.40 \pm 0.03}$ | $21.99 \pm 0.02$ | 2.68% |
| 0.75 | $\mathbf{18.25 \pm 0.01}$ | $18.91 \pm 0.08$ | 3.49% |
| 0.7 | $\mathbf{16.09 \pm 0.02}$ | $16.32 \pm 0.01$ | 1.41% |

**Training details.** We follow the settings from Geifman & El-Yaniv (2019) whenever possible. We train each model for 300 epochs, with an initial learning rate of 0.1 which is reduced by half every 25 epochs. The model is optimized using stochastic gradient descent (SGD) with momentum 0.9 and weight decay $5e-4$. All experiments are repeated 3 times similar to SELECTIVENET. We report both the average value and standard deviation (following $\pm$). The classification error rate that we report in our experiments is the percentage of incorrectly predicted examples among all the accepted examples, by evaluating the selective model on the hold-out test sets. Note that classification error rate is the same as empirical selective risk value when using the 0-1 loss. All experiments are run using NVIDIA V100 GPUs.

**Post-training calibration.** We report the calibrated selective error rate as the metric for comparing against the baseline for each target coverage value. To get the calibrated selective error, we estimate an appropriate threshold $\tau$ for selection head output values on a validation set, and use the following calibrated rule for our predictions:

$$(f, g)(x) = \begin{cases} f(x), & \text{if } g(x) \geq \tau; \\ \text{ABSTAIN}, & \text{otherwise .} \end{cases} \tag{8}$$

This calibration process is used to correct for the difference between the post-training empirical coverage $\hat{\phi}$ and the target coverage $c$. Specifically, given a validation set $V$, we set $\tau$ to be the $100(1-c)$ percentile of the distribution of $g(x_i), x_i \in V$.

## 5.2 SELECTIVE CLASSIFICATION RESULTS

**Moderate-sized CNN.** As with Hacohen & Weinshall (2019), we conduct experiments using moderate-sized convolutional neural networks (CNN) on all three datasets CIFAR10, CIFAR100 and SVHN. The corresponding results are shown in Table 1, Table 2, and Table 3. It can be seen that, as the target coverage rate becomes smaller, the error rate is lower for both our method and the baseline. This behavior is expected because, in general, any selective model should be able to trade-off coverage for error rate, i.e., lower the error by reducing coverage. However, achieving lower error becomes more challenging when the target coverage is smaller because the model needs to figure out which additional examples should be further rejected.

Overall, our method outperforms the baseline by achieving lower selective error rates for each target coverage rate (corresponding to each row in the above mentioned tables). Our method consistently

Table 5: Ablation study with moderate-sized CNN on CIFAR10. Testing classification error rates (in %) are shown below.

| Coverage | Our Method | SelectiveNet | Inception | Vanilla-Selection |
|----------|------------|--------------|-----------|-------------------|
| 0.9 | $\mathbf{5.63 \pm 0.04}$ | $6.07 \pm 0.1$ | $5.87 \pm 0.05$ | $5.98 \pm 0.04$ |
| 0.8 | $\mathbf{3.02 \pm 0.02}$ | $3.47 \pm 0.07$ | $3.14 \pm 0.05$ | $3.35 \pm 0.04$ |
| 0.7 | $\mathbf{1.47 \pm 0.02}$ | $1.67 \pm 0.08$ | $1.59 \pm 0.02$ | $1.66 \pm 0.03$ |

improves upon the baseline across all three datasets. These improvements are even higher for CIFAR10 and SVHN (up to $10\%$), especially when the target coverage is small. This demonstrates that our method is very effective at trading off the coverage for even lower error rate.

**VGG.** We also conduct experiments using the VGG-16 model (Simonyan & Zisserman, 2014) on the CIFAR100 dataset, because of VGG-16's strong prediction ability which is desired for the CIFAR100 dataset. VGG is a larger image recognition model with more layers. As shown in Table 5, our method still outperform the baseline SELECTIVENET consistently across all different target coverage rates. This verifies that our method can be applied to different backbone models. It is worth noting that, compared with the moderate-sized CNN on CIFAR100 shown in Table 2, the error rate for either method is smaller smaller in Table 5 for the same coverage rate. This is expected because VGG model is larger in size (more layers). In addition, the average improvement from Table 2 is $1.61\%$ while the average improvement from Table 5 is $2.07\%$. This implies that our method can achieve an even greater improvement with larger models.

## 5.3 ABLATION STUDY

We conduct ablation study to validate our design decisions by comparing our method against two variations of it. Note that our method is composed of two components: (1) estimating the difficulty scores, and (2) selecting examples for each mini-batch during training. To show that both components play an important role in the overall success of our method, we implement two variants, each using one component from existing work. **Inception** is a variant of our method using the pre-trained inception model for difficulty estimation (i.e., *transfer scoring function* in Hacohen & Weinshall (2019)). **Vanilla-Selection** is another variant of our method using the vanilla method of selecting examples for each mini-batch that is typically used in existing curriculum literature. Different from our method which considers ordering at the mini-batch level, existing curriculum learning literature typically consider ordering at the example level while selecting examples for each mini-batch. Specifically, they construct the mini-batch by uniformly sampling from a working subset, which starts with a few easiest examples, and is being appended with harder and harder examples after every several epochs until all the training examples are added (Wang et al., 2021; Soviany et al., 2022). We adopt this strategy in **Vanilla-Selection** which can be viewed as applying the vanilla curriculum learning (e.g., Algorithm 1 from Hacohen & Weinshall (2019)) to selective classification domain.

As reported in Table 5, both variants have higher error rates than our method, validating that both components are integral to and contribute to our method's impressive performance. Especially, applying the vanilla curriculum learning to the selective classification domain as in **Vanilla-Selection** is not as good as our method which has a specially tailored curriculum learning strategy. In addition, even though our method involves an overhead of model training in order to get the difficulty score estimation, this overhead is paid only once even if the coverage needs to change. **Inception** is the variant without this overhead since the off-the-self pre-trained inception model can be readily obtained online. If one does not wish to pay the upfront overhead of our method, these results imply that even **Inception** outperforms the baseline.

## 6 CONCLUSION

In this paper, we proposed a curriculum-inspired method to train selective neural network models by leveraging difficulty scores. Our method calibrates the ratio of easy and hard examples in each mini-batch to match what is best desired by the training objective. We generalize curriculum learning by utilizing difficulty scores to reorder mini-batches rather than simply reorder examples. Extensive experiments show the superior performance of our method compared to state-of-the-art SELECTIVENET.

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

# A APPENDIX

## A.1 SCENARIOS WHERE SELECTIVE CLASSIFICATION IS NEEDED

In a nutshell, selective classification is needed in settings where prediction error of the model can cause critical consequences (e.g., in medical diagnosis or self-driving cars). In these settings, we would like the model to proactively indicate when it's not confident in its prediction, so that human intervention is triggered.

**Medical diagnosis.** In medical diagnosis, we can train a machine learning model to predict if a patient has a tumor based on the MRI image. When the model is not confident enough, we can ask an oncologist, and even call upon an urgent meeting of multiple oncologists across different hospitals for a group consultation (Kompa et al., 2021).

**Self-driving cars.** In self-driving cars, we can train a machine learning model to detect the traffic light, which will help the self-driving car to decide what to do at cross-roads. However, when the model is not confident enough (e.g., traffic light is not clear due to rain or obstruction), the human driver should be alerted and take control of the car.

## A.2 DETAILED EXPLANATIONS ABOUT TARGET COVERAGE MATCH

We elaborate on why it's ideal to match the proportion of easy examples in each mini-batch with the target coverage as briefly mentioned in Section 4.1. We start by showing that the empirical coverage $\hat{\phi}$ will converge to the target coverage $c$. Then we show that a mini-batch in which the proportion

of easy examples is the same as the target coverage is ideal. Therefore, a mini-batch which has the proportion of easy examples different from the target coverage will misguide the selection function regarding which examples should be accepted or rejected, which further affects the prediction head's learning progress.

**Empirical coverage converges to target coverage.** As a result of minimizing Equation 5, the empirical coverage $\hat{\phi}$ will converge to the target coverage $c$. Without being too technically rigorous, we demonstrate this by showing that, whether or not the empirical coverage $\hat{\phi}$ is larger than target coverage $c$, the loss $\mathcal{L}_{(f,g)}$ (i.e., left hand side of Equation 5) will become smaller if empirical coverage is moved closer to target coverage.

- When the empirical coverage is larger than the target coverage, the second term of Equation 5 becomes $0$. Note that the first term $\hat{r}$ is the average loss of accepted examples. By simply rejecting the example with the highest loss (i.e., decreasing the empirical coverage), the first term $\hat{r}$ will become smaller while keeping the second term still $0$. Therefore, when the empirical coverage is larger than the target coverage, decreasing the empirical coverage will make the loss $\mathcal{L}_{(f,g)}$ smaller.

- When the empirical coverage is smaller than the target coverage, the second term is a positive value. As long as the hyperparameter $\lambda$ is large enough to make the second term dominant, the loss $\mathcal{L}_{(f,g)}$ will become smaller by increasing the empirical coverage. As suggested in Geifman & El-Yaniv (2019), the hyperparameter $\lambda$ is chosen to be a large value 32 in our experiments. Therefore, when the empirical coverage is smaller than the target coverage, increasing the empirical coverage will make the loss $\mathcal{L}_{(f,g)}$ smaller.

**Mini-batch should ideally match target coverage.** We now show that a mini-batch in which the proportion of easy examples is the same as the target coverage is ideal. We demonstrate it by showing that, when the mini-batch has the proportion of easy examples different from the target coverage, as a result of minimizing the loss during the training, even an already perfect selection function tends to move to the direction where mistakes will be made (i.e., reject easy examples or accept hard examples). Note that, by an already perfect selection function, we mean that the selection function accepts or rejects examples based on the ground truth (i.e., reject hard examples or accept easy examples).

- When the proportion of easy examples in the mini-batch is larger than the target coverage, a perfect selection function gives the empirical coverage on this mini-batch that is larger than target coverage. As we have just discussed above, when the empirical coverage is larger than the target coverage, decreasing the empirical coverage will make the loss $\mathcal{L}_{(f,g)}$ smaller. It implies that the selection function tends to reject more examples, which are actually accepted by a perfect selection function.

- When the proportion of easy examples in the mini-batch is smaller than the target coverage, a perfect selection function gives the empirical coverage on this mini-batch that is smaller than target coverage. As we have just discussed above, when the empirical coverage is smaller than the target coverage, increasing the empirical coverage will make the loss $\mathcal{L}_{(f,g)}$ smaller. It implies that the selection function tends to accept more examples, which are actually rejected by a perfect selection function.

## A.3    MORE DETAILS ABOUT EXPERIMENTAL SETUP

The moderate-sized convolutional network architecture contains $8$ convolutional layers with $32, 32, 64, 64, 128, 128, 256, 256$ filters respectively. The filter size is $3 \times 3$ for the first 6 layers and $2 \times 2$ for the last 2 layers. There is a $2 \times 2$ max-pooling layer and a dropout layer with rate $0.25$ after every two layers. Following the convolutional layers, there is a fully-connected layer with $512$ units and a dropout layer with rate $0.25$. At last, another fully connected layer is used to match the number of classes in the dataset. We omit the detailed layer-by-layer description of VGG-16 as it is described in Simonyan & Zisserman (2014). In addition, the prediction head $f$ and auxiliary head $h$ are fully-connected softmax layers, and the selection head $g$ is a fully-connected hidden layer with 513 neurons, followed by batch normalization, ReLU activation and another fully connected layer to one output neuron with a sigmoid activation. These architectures have been used in the

Table 6: Ablation study with moderate-sized CNN on CIFAR10. Testing error rates (in %) are shown below.

| Coverage | Our Method | SelectiveNet | Inception | Vanilla-Selection |
|---|---|---|---|---|
| 0.95 | **7.32 ± 0.05** | 7.57 ± 0.1 | 7.43 ± 0.07 | 7.51 ± 0.04 |
| 0.9 | **5.63 ± 0.04** | 6.07 ± 0.1 | 5.87 ± 0.05 | 5.98 ± 0.04 |
| 0.85 | **4.30 ± 0.04** | 4.53 ± 0.08 | 4.37 ± 0.03 | 4.42 ± 0.02 |
| 0.8 | **3.02 ± 0.02** | 3.47 ± 0.07 | 3.14 ± 0.05 | 3.35 ± 0.04 |
| 0.75 | **2.16 ± 0.04** | 2.45 ± 0.07 | 2.23 ± 0.04 | 2.38 ± 0.09 |
| 0.7 | **1.47 ± 0.02** | 1.67 ± 0.08 | 1.59 ± 0.02 | 1.66 ± 0.03 |

recent selective classification and curriculum learning papers (Geifman & El-Yaniv, 2019; Hacohen & Weinshall, 2019; Wu et al., 2020).

### A.4 MORE EXPERIMENTAL RESULTS

The complete results of ablation study are shown in Table 6.

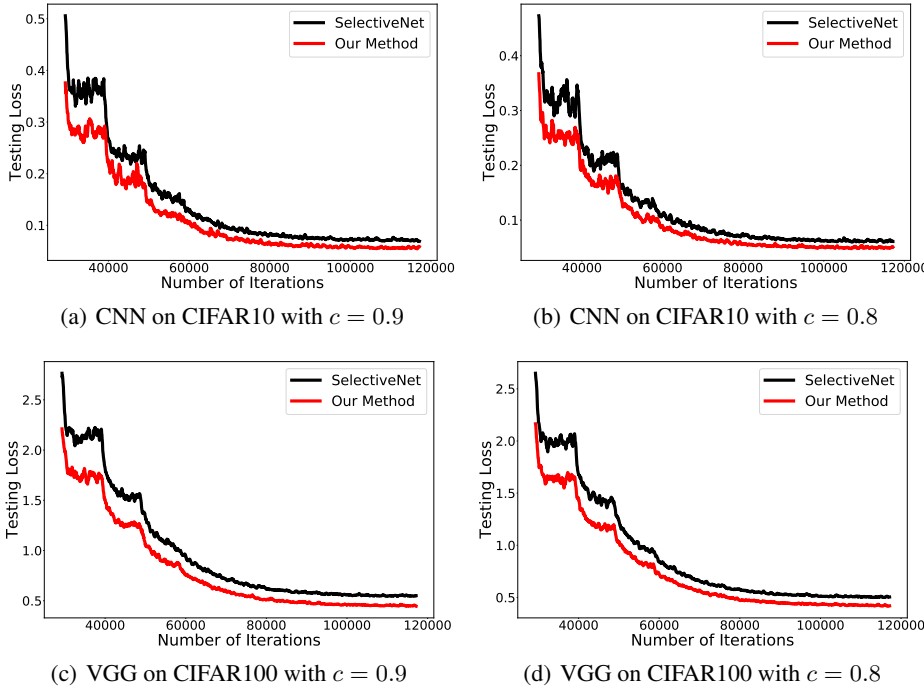

(a) CNN on CIFAR10 with $c = 0.9$

(b) CNN on CIFAR10 with $c = 0.8$

(c) VGG on CIFAR100 with $c = 0.9$

(d) VGG on CIFAR100 with $c = 0.8$

Figure 2: Testing loss vs. the number of iterations for different models on different datasets. Our method convergences faster by achieving a lower testing loss for the same number of iterations.

### A.5 CONVERGENCE SPEED

It has been shown that curriculum learning can improve the convergence speed when training on various classical neural network models (Hacohen & Weinshall, 2019; Li et al., 2021). In this section, we investigate whether this applies to our method as well. We plot the curves of testing loss versus the number of iterations under different settings in Figure 2. Specifically, Figures 2 (a) and (b) correspond to training moderate-sized CNN on the CIFAR10 dataset with a target coverage of $c = 0.9$ and $c = 0.8$, respectively, while Figures 2 (c) and (d) correspond to training VGG-16 on the CIFAR100 dataset with a target coverage of $c = 0.9$ and $c = 0.8$, respectively. We can see that our method has a better convergence rate than the baseline because our method achieves a lower testing loss for the same number of training iterations than SELECTIVENET. To better understand how faster our method's converges, we also measure the number of iterations to reach the same testing loss as summarized in Table 7. Specifically, the testing loss to reach is set as the lowest testing loss

Table 7: The number of iterations to reach the target loss for training moderate-sized CNN on the CIFAR10 dataset. The target loss value is the lowest testing loss achieved by SELECTIVENET for each coverage rate.

| Coverage | # of Iterations | | Impr. |
| | Our Method | SelectiveNet | |
|---|---|---|---|
| 0.95 | 332k | 694k | 52% |
| 0.9 | 73k | 116k | 37% |
| 0.85 | 97k | 459k | 79% |
| 0.8 | 74k | 115k | 36% |
| 0.75 | 75k | 109k | 31% |

Table 8: Coverage violation of moderate-sized CNN on CIFAR10. Empirical Coverage (in %) are shown for different target coverage rates. Our method achieves smaller coverage violation (difference between target coverage and empirical coverage) than SELECTIVENET.

| Coverage (in %) | Our Method | | SelectiveNet | | Violation Reduction |
| | Empirical Coverage | Violation | Empirical Coverage | Violation | |
|---|---|---|---|---|---|
| 95 | $90.2 \pm 0.1$ | 4.8 | $90.0 \pm 0.1$ | 5.0 | 4% |
| 90 | $84.7 \pm 0.1$ | 5.3 | $84.1 \pm 0.2$ | 5.9 | 10.2% |
| 85 | $80.7 \pm 0.2$ | 4.3 | $80.2 \pm 0.1$ | 4.8 | 10.4% |
| 80 | $76.6 \pm 0.1$ | 3.4 | $76.3 \pm 0.2$ | 3.7 | 8.1% |
| 75 | $73.1 \pm 0.1$ | 1.9 | $72.7 \pm 0.3$ | 2.3 | 17.4% |
| 70 | $69.3 \pm 0.1$ | 0.7 | $68.8 \pm 0.1$ | 1.2 | 41.7% |

achieved by SELECTIVENET. It can be seen that our method can reduce the the number of iterations by up to 80%. This demonstrates that our method is significantly faster than the baseline during training.

## A.6  COVERAGE VIOLATION

Our method leverages the difficulty scores to calibrate the ratio of the difficult and easy examples in each mini-batch. This could effectively reduce the discrepancy between the target coverage and empirical coverage (i.e., the coverage on the testing set before applying the calibration process as described in Equation 8). We measure the empirical coverage and how much it differs from the target coverage (i.e., violation) as summarized in Table 8. As expected, the empirical coverage achieved by our method is closer to the target coverage, resulting in smaller violation compared to the baseline (i.e., SELECTIVENET). The relative reduction of the violation achieved by our method is larger than 10% for most values of target coverage (i.e., 90%, 70%). Especially, when the target coverage is small, i.e., 70%, the violation reduction is as high as 41.7%.

