# OpenReview forum: "Curriculum-inspired Training for Selective Neural Networks"
_ICLR.cc/2023/Conference — Submitted to ICLR 2023_

### Official Review · Reviewer_F3no · 2022-10-24

**Confidence:** 3
**Clarity, Quality, Novelty And Reproducibility:** <Covered in comments above.>
**Correctness:** 4
**Technical Novelty And Significance:** 2
**Empirical Novelty And Significance:** 3
**Recommendation:** 6

**Strength And Weaknesses:**

Comments:
1. First paper to leverage the difficulty scores obtained by selective classification in curriculum learning framework.
2. The contribution of this work is (1) defining the coverage term in the loss function. (2) use of mini-batch learning for curriculum inspired training.
3. The experiments demonstrate the effectiveness of their technique over SelectiveNet.
4. Have you tried different sampling strategies, instead of uniformly distributing easy and difficult samples in a mini-batch? Something like importance sampling based on the difficulty scores?
5. Any theoretical guarantees of using this type of curriculum-inspired training?
6. Any experiments/datasets where this technique failed to improve over vanilla learning?
7. Any observations on getting worse results for this way of obtaining mini-batches? Some theoretical guarantees on convergence will be perfect but even empirical evaluation will be good.

This work is clearly presented and well written. The appendix also covers more details on ablation studies and convergence rates. I feel that this work is incremental and not very novel but useful nonetheless.


**Summary Of The Paper:**

This paper deals with NN models that are allowed to abstain from making a prediction whenever it is not sufficiently confident. These selective models are used to minimize risk of wrong prediction. Their model is based on SelectiveNet, which has 3 heads, one for minimizing the error for selected examples, one for deciding whether to select or reject examples and the last one for minimizing overall error rate. Given an order set of examples based on their difficulty scores, the curriculum-inspired training proposed by this work samples a mini-batch consisting of easy and hard examples to iteratively improve coverage and prediction accuracy of the NN model. They show improved performance of their technique over SelectiveNet.


**Summary Of The Review:**

This work is clearly presented and well written. The appendix also covers more details on ablation studies and convergence rates. I feel that this work is not very novel but useful nonetheless.

---

> ### Author Response · Authors · 2022-11-19
> **author response**
>
> We thank the reviewer for the supportive comments. Below we address all the major comments, numbered as **C1** to **C4** (corresponding answers are numbered as **A1**-**A4**)
>
> **C1**: Have you tried different sampling strategies, instead of uniformly distributing easy and difficult samples in a mini-batch? Something like importance sampling based on the difficulty scores?
> **A1**: We thank the reviewer for mentioning this. As with the existing curriculum learning idea [1], we do not consider importance sampling within each mini-batch. Though we believe importance sampling is an interesting line of research, it is unclear what is the best way to combine them. Nevertheless, we think it is an interesting direction for future work.
>
> **C2**: Any theoretical guarantees of using this type of curriculum-inspired training?
> **A2**: As with our baseline [2], we focus on empirical performance. We agree that a theoretical analysis would be interesting. However, it is hard to analyze curriculum-inspired training especially because of the interplay between three network heads (i.e., prediction, selection and auxiliary) during training.
>
> **C3**: Any experiments/datasets where this technique failed to improve over vanilla learning?
> **A3**: We thank the reviewer for this great question. The curriculum learning idea may not be beneficial when there is not much difference between example difficulties. For example, for some datasets, examples may be approximately equally hard or easy. In these cases, as with curriculum learning, we may not observe improvement for our technique.
>
> **C4**: Any observations on getting worse results for this way of obtaining mini-batches? Some theoretical guarantees on convergence will be perfect but even empirical evaluation will be good.
> **A4**: We have not observed getting worse results for this way of obtaining mini-batches. However, we do agree that our technique has some limitations just as with other curriculum learning methods (kindly refer to A3 as above). Please refer to A2 for our answer towards theoretical analysis.
>
> **Reference**.
> [1] Hacohen, Guy, and Daphna Weinshall. "On the power of curriculum learning in training deep networks." International Conference on Machine Learning. PMLR, 2019.
> [2] Geifman, Yonatan, and Ran El-Yaniv. "Selectivenet: A deep neural network with an integrated reject option." International conference on machine learning. PMLR, 2019.

---

### Official Review · Reviewer_oBun · 2022-10-24

**Confidence:** 5
**Clarity, Quality, Novelty And Reproducibility:** The paper is clear, but the novelty i…
**Correctness:** 2
**Technical Novelty And Significance:** 2
**Empirical Novelty And Significance:** 2
**Recommendation:** 5

**Strength And Weaknesses:**

Strengths:
- The paper is well-written and easy to follow.

- The authors study an interesting topic.

Weaknesses:
- The level of novelty is limited in my opinion. The curriculum learning method is quite standard. The novelty consists in applying curriculum learning to selective nets, which does not represent a significant contribution.

- The authors use the relative improvements in terms of error rate, which is deceiving. The relative gains in terms of accuracy would be very different. Nonetheless, the absolute gains seem rather small.

- The authors use only small-scale and low-resolution datasets. It is thus unclear if the method would help on large-scale datasets where there are more examples per class available.

- The authors should also relate and compare with curriculum methods not using example difficulty scores, e.g. [A].

- In my opinion, the studied problem is related to the idea of dispatching test examples to different models based on image difficulty [B]. It would be nice for the authors to mention that hard examples could be processed by another (more confident) model instead of rejecting them.

[A] Samarth Sinha, Animesh Garg, and Hugo Larochelle, “Curriculum by smoothing.” In Neural Information Processing Systems, pp. 21653-21664, 2020.

[B] Petru Soviany, and Radu Tudor Ionescu. "Optimizing the trade-off between single-stage and two-stage deep object detectors using image difficulty prediction." In International Symposium on Symbolic and Numeric Algorithms for Scientific Computing, pp. 209-214, 2018.


**Summary Of The Paper:**

The paper studies curriculum learning for selective neural networks. These are models that are allowed to reject examples if the confidence is low. The method proposed by the authors relies on carefully selecting the easy vs hard examples to be included in each mini-batch. Experiments are reported on CIFAR10, CIFAR100 and SVHN, comparing the proposed method with vanilla curriculum and non-curriculum baselines.

**Summary Of The Review:**

My main concerns are related to the low-level of novelty and the rather small improvements on low-resolution datasets.

---

> ### Author Response · Authors · 2022-11-19
> **author response**
>
> We thank the reviewer for the insightful comments. Below we address all the major comments, numbered as **C1** to **C4** (corresponding answers are numbered as **A1**-**A4**)
>
>
> **C1**: The level of novelty is limited in my opinion. The curriculum learning method is quite standard. The novelty consists in applying curriculum learning to selective nets, which does not represent a significant contribution.
> **A1**: We would like to kindly note that, as we explained in Section 2,  “there is a huge surge of interests in applying curriculum learning to different domains in recent years, e.g., image classification (Guo et al., 2018), machine translation (Platanios et al., 2019), language model pre-training (Li et al., 2021), healthcare prediction (El-Bouri et al., 2020) and graph learning (Gong et al., 2019).” We are the first to apply curriculum learning to selective nets. In addition, we improve the vanilla curriculum learning idea by (1) calibrating the ratio of easy and hard examples to mitigate the misguiding noise issue, (2) exploiting the difficulty ordering at the mini-batch level to improve the accuracy and convergence speed. Considering all these factors, we believe that our paper is a valuable and significant progress on both fields of curriculum learning and selective classification.
>
> **C2**: The authors use the relative improvements in terms of error rate, which is deceiving. The relative gains in terms of accuracy would be very different. Nonetheless, the absolute gains seem rather small.
> **A2**: We would like to note that relative improvements are also used in our baseline paper (e.g., Table 2 in [1]). In addition, note that in many experimental settings, the error rates are already very low, which makes even a small improvement hard to achieve. Actually, the improvements shown in other papers are also in a similar range (for example see Table 2 in SelectiveNet [1], Table 4 in [2], and Table 4 in [3]).
>
> **C3**: The authors use only small-scale and low-resolution datasets. It is thus unclear if the method would help on large-scale datasets where there are more examples per class available.
> **A3**: We understand that there are larger and high-resolution datasets. Though it’s great to include more datasets, note that the experiments for selective classification are time consuming, because we need to retrain the models for each coverage value, and each experiment is repeated $3$ times with different random seeds. What’s more, as far as we know, we have used more difficult datasets compared to other selective classification papers [1,2,3]: CIFAR100 is used in our experiments.
>
>
> **C4**: The authors should also relate and compare with curriculum methods not using example difficulty scores, e.g. [A]. In my opinion, the studied problem is related to the idea of dispatching test examples to different models based on image difficulty [B]. It would be nice for the authors to mention that hard examples could be processed by another (more confident) model instead of rejecting them.
> **A4**: We thank the reviewer for mentioning these interesting works. We are happy to add more discussions about them in our paper.
>
> **Reference**.
> [1] Geifman, Yonatan, and Ran El-Yaniv. "Selectivenet: A deep neural network with an integrated reject option." International conference on machine learning. PMLR, 2019.
> [2]  Liu, Ziyin, et al. "Deep gamblers: Learning to abstain with portfolio theory." Advances in Neural Information Processing Systems 32 (2019).
> [3] Huang, Lang, Chao Zhang, and Hongyang Zhang. "Self-adaptive training: beyond empirical risk minimization." Advances in neural information processing systems 33 (2020): 19365-19376.

---

### Official Review · Reviewer_bj8f · 2022-10-24

**Confidence:** 3
**Correctness:** 3
**Technical Novelty And Significance:** 2
**Empirical Novelty And Significance:** 2
**Recommendation:** 3

**Clarity, Quality, Novelty And Reproducibility:**

This paper is well-written and reproducible. The ideas are new for selective neural networks but are very straightforward, so only contains modest novelty.

**Strength And Weaknesses:**

The proposed idea is straightforward and makes sense to me. But I feel the empirical evaluation is not rigorous enough to validate the usefulness of the proposed method. Here are my concerns:

(1) The improvement over the baseline SelectiveNet is quite marginal, e.g., ~0.5% error rate reduction on CIFAR-100 in Table 2.

(2) There are no results on ImageNet. It is strongly encouraged to show ImageNet results for image classification.

(3) There lacks diversity in network architectures. This work only shows results on a small CNN and VGG-16, both of which are probably far from the SOTA network architectures in computer vision, e.g., EfficientNet, FBNet, etc.

(4) An existing work (“Wisdom of Committees: An Overlooked Approach To Faster and More Accurate Models” ICLR 2022) shows that a couple of simple of metrics (e.g., maximum confidence, entropy, etc) can be used to make rejection decisions as well (although they use it in the context of model cascades). Due to the simplicity of those metrics, a comparison against them is necessary and we should expect an obvious improvement to justify the usefulness of the proposed method.


**Summary Of The Paper:**

This work focuses on the training of selective networks, where the network knows then to reject to answer for unconfident examples. They propose a curriculum-inspired method to train selective neural works, where they (1) consider the target converge ratio when sampling batches, and (2) use the example difficulty score to to design the curriculum. They conduct experiments on CIFAR-10/100 and SVHN to demonstrate the usefulness of their method.


**Summary Of The Review:**

The novelty of this work is not significant as the main ideas in this work are straightforward. In this case, I would like to look for rigorous empirical evaluation. At this point, I feel a ton of experiments need to be added to fully justify the usefulness of the proposed. The current performance improvement is marginal.

---

> ### Author Response · Authors · 2022-11-19
> **author response**
>
> We thank the reviewer for the insightful comments. Below we address all the major comments, numbered as **C1** to **C3** (corresponding answers are numbered as **A1**-**A3**)
>
> **C1**: The improvement over the baseline SelectiveNet is quite marginal, e.g., ~0.5% error rate reduction on CIFAR-100 in Table 2.
> **A1**: We would like to note that in many experimental settings, the error rates are already very low, which makes even a small improvement hard to achieve, thus deemed significant. Actually, the improvements shown in other papers are also in a similar range (for example see Table 2 in SelectiveNet [1]).
>
> **C2**: There are no results on ImageNet. It is strongly encouraged to show ImageNet results for image classification. There lacks diversity in network architectures. This work only shows results on a small CNN and VGG-16, both of which are probably far from the SOTA network architectures in computer vision, e.g., EfficientNet, FBNet, etc.
> **A2**: We thank the reviewer for bringing this up. We understand that there are larger datasets and better network architectures. However, we would like to note that we chose those datasets and network architectures in our paper according to what are typically used in other recent papers for selective classification [1,2,3]. Though it’s great to include more datasets and architectures, note that the experiments for selective classification are time consuming, because we need to retrain the models for each coverage value, and each experiment is repeated $3$ times with different random seeds. What’s more, as far as we know, we have used more difficult datasets compared to other selective classification papers [1,2,3]: CIFAR100 is used in our experiments.
>
> **C3**: An existing work (“Wisdom of Committees: An Overlooked Approach To Faster and More Accurate Models” ICLR 2022) shows that a couple of simple of metrics (e.g., maximum confidence, entropy, etc) can be used to make rejection decisions as well (although they use it in the context of model cascades). Due to the simplicity of those metrics, a comparison against them is necessary and we should expect an obvious improvement to justify the usefulness of the proposed method.
> **A3**: We thank the reviewer for mentioning this interesting work. We are happy to add more discussions about it in our paper.
>
> **Reference**.
> [1] Geifman, Yonatan, and Ran El-Yaniv. "Selectivenet: A deep neural network with an integrated reject option." International conference on machine learning. PMLR, 2019.
> [2]  Liu, Ziyin, et al. "Deep gamblers: Learning to abstain with portfolio theory." Advances in Neural Information Processing Systems 32 (2019).
> [3] Huang, Lang, Chao Zhang, and Hongyang Zhang. "Self-adaptive training: beyond empirical risk minimization." Advances in neural information processing systems 33 (2020): 19365-19376.

---

> > ### Comment · Reviewer_bj8f · 2022-11-21
> > **After rebuttal**
> >
> > Thanks to the authors for the response.
> >
> > Regarding C1 & C2, I agree that the error rate somewhat saturates on CIFAR-10/100. But that's exactly why it's important to evaluate on larger and harder benchmarks, e.g., ImageNet, to better compare the proposed method against baselines. Unfortunately, the evaluation is still missing.
> >
> > Regarding C3, discussing those simple metrics in the text is not enough and there should be clear empirical results to show the improvement of the proposed method against those simple metrics, which are still missing.
> >
> > Therefore,  the rebuttal failed to resolve my concerns so I decided to decrease the rating to Reject.

---

### Official Review · Reviewer_xFiA · 2022-10-24

**Confidence:** 4
**Correctness:** 4
**Technical Novelty And Significance:** 2
**Empirical Novelty And Significance:** 2
**Recommendation:** 5

**Clarity, Quality, Novelty And Reproducibility:**

Paper is written in a simple manner and it is easy to follow. This work is indeed novel in that in integrates curriculum learning while training selective neural networks. It does not simply follow vanilla curriculum strategy but creates a mini-batch that respects the coverage constraint.


**Strength And Weaknesses:**


Strengths:
- Incorporating Curriculum learning in Selective Classification is indeed novel
- Proposed carefully designed mini-batch design is indeed relevant to the coverage constraint being tackled in the Selective Classification

Weaknesses:
- Poor empirical performance (in many cases the improvement in error is in the range of 0.3-0.5 see Table~1, 2)
- Lack of ablations that justify the design choices (selectivenet over other abstaining neural networks, pacing function, scoring function)
- Does not address the computational overhead of the proposed method
- It is unclear why this work chooses the proposed difficulty score estimator. It is by no means computational cheap as compared to simpler metrics such as entropy, softmax margin, etc.

Questions for Authors:

- Choice of pacing function : For completeness, could you describe what the fixed exponential pacing function is? Did you try some other pacing functions? Since this problem ( CL + abstaining classifier ) is different than existing CL work, it is unclear why fixed exponential pacing function should be the de-fact choice in this scenario?

- Choice of scoring function : Outlier detection algorithm ( Minimum Covariance Determinant (MCD) ) seems like an expensive way to measure the difficulty of examples. Is there a comparison on the training time split, i.e., how much time gets split between training the auxiliary classifier, how much goes in computing the MCD scores, how much in training the abstaining classifier? Have you tried other simpler scoring functions such as entropy, softmax margin, etc.?

- How often is the MCD invoked? Are the scores refreshed every epoch in the training or only once in a while?

- Could you elaborate the point regarding the use of inception network w.r.t. MCD?

- Could you shed some lights as to why the gains are not impressive in Table 1 and 2?

- Can you plot the hardness measure to see how well separated easy/hard examples are for various coverage levels?

-  Is there a reason why SelectiveNet was used as the abstaining architecture? There are other superior networks than SelectiveNet, for instance,
     - DeepGamblers : https://proceedings.neurips.cc/paper/2019/file/0c4b1eeb45c90b52bfb9d07943d855ab-Paper.pdf
     - One-Sided Predictions : http://proceedings.mlr.press/v130/gangrade21a/gangrade21a.pdf
     - Self-Adaptive Training : https://proceedings.neurips.cc/paper/2020/file/e0ab531ec312161511493b002f9be2ee-Paper.pdf
     - NNTD : https://arxiv.org/pdf/2205.13532.pdf




**Summary Of The Paper:**

This work introduces curriculum learning in training selective neural networks.
- Selective Neural Network, also referred to as the abstaining classifier, have the option to reject an input, i.e., abstain from prediction. Typically, this abstention mechanism is learnt to be a measure of the difficulty of the input.
- Curriculum Learning trains a standard neural network by sorting the inputs according to some difficulty / uncertainity measure and serving these data points in the order from easy to hard during training.

This work merges the above two ideas. Specifically, it trains a selective neural network using curriculum learning. Although one could leverage the vanilla curriculum learning strategy, the notion of rejection provides an opportunity to align the difficulty notion used in the curriulum learning and the selective neural networks. This work calibrates the input into easy and hard examples and serves each mini-batch to the abstaining network such that it can enforce a coverage constratint ( amount of examples that the network accepts for prediction ). Empirical evaluation shows that the proposed curriculum learning design for selective neural networks is better than vanilla techniques.


**Summary Of The Review:**

This work is indeed novel in that in integrates curriculum learning while training selective neural networks. It does not simply follow vanilla curriculum strategy but creates a mini-batch that respects the coverage constraint.

I had expected a bit more thoughts in the design choices such as scoring and pacing functions. Since these are very crucial to the success of proposed method. In addition, choice of selective net is also unclear. There are other abstaining classifiers such as DeepGamblers and NNTD with better performance. I would have expected a few ablations that justify the choice of selectivenet.

---

> ### Author Response · Authors · 2022-11-19
> **author response**
>
> We thank the reviewer for the insightful comments. Below we address all the major comments, numbered as **C1** to **C9** (corresponding answers are numbered as **A1**-**A9**).
>
> **C1**: Poor empirical performance (in many cases the improvement in error is in the range of 0.3-0.5 see Table~1, 2).
> **A1**: We would like to note that in many experimental settings, the error rates are already very low (< 10\%), which makes even a small improvement hard to achieve, thus deemed significant. Actually, the improvements shown in other papers are also in the range of 0.3 - 0.5 (for example see Table 2 in SelectiveNet [1]).
>
> **C2**: Lack of ablations that justify the design choices (selectivenet over other abstaining neural networks, pacing function, scoring function).
> **A2**: We thank the reviewer for pointing this out. As we have explained in Section 4.2, we only consider the fixed exponential pacing function because it has the right balance between good performance and the number of additional hyperparameters [2]. In addition, we have discussed other scoring functions in our ablation study (see **Inception** in Section 5.3).
>
> **C3**: It is unclear why this work chooses the proposed difficulty score estimator. It is by no means computational cheap as compared to simpler metrics such as entropy, softmax margin, etc.
> **A3**: Our empirical results show that the proposed difficulty score estimator leads to good performance. Actually, in our ablation study, we have considered an alternative difficulty score estimator (see **Inception** in Section 5.3), which has worse performance than the proposed one.
>
> **C4**: Choice of pacing function : For completeness, could you describe what the fixed exponential pacing function is? Did you try some other pacing functions? Since this problem ( CL + abstaining classifier ) is different than existing CL work, it is unclear why fixed exponential pacing function should be the de-fact choice in this scenario?
> **A4**: We thank the reviewer for bringing this up. We didn’t elaborate on the exponential pacing function because it’s just a small piece of our method. A very good illustration is shown in Figure 1 from [2] (the red curve in Figure 1 of [2] is the exponential pacing function which is what we used in our paper). We are happy to have more discussion on pacing function in our appendix. Please refer to A2 for more information around why fixed exponential pacing function is chosen.
>
> **C5**: Choice of scoring function : Outlier detection algorithm ( Minimum Covariance Determinant (MCD) ) seems like an expensive way to measure the difficulty of examples. Is there a comparison on the training time split, i.e., how much time gets split between training the auxiliary classifier, how much goes in computing the MCD scores, how much in training the abstaining classifier? Have you tried other simpler scoring functions such as entropy, softmax margin, etc.?
> **A5**: As we have mentioned in Section 4.3, we pick MCD because of its robustness to noise and outliers. Please also refer to A3.
>
> **C6**: How often is the MCD invoked? Are the scores refreshed every epoch in the training or only once in a while?
> **A6**: Thanks for the question. MCD is only invoked once after the pre-training is finished, in order to compute the distance in the hidden feature space. We have a detailed explanation about this in Section 4.3.
>
> **C7**: Could you elaborate the point regarding the use of inception network w.r.t. MCD?
> **A7**: We would like to clarify that **inception** method in Section 5.3 is an alternative method that we have considered for computing the difficulty scores. It does not involve MCD in any way.
>
> **C8**: Could you shed some lights as to why the gains are not impressive in Table 1 and 2?
> **A8**: This is because the error rates are already very low in Table 1 and 2. Therefore, a small improvement is hard to achieve, thus deemed impressive. Please kindly refer to A1 as well.
>
> **C9**: Is there a reason why SelectiveNet was used as the abstaining architecture? There are other abstaining classifiers such as DeepGamblers and NNTD with better performance.
> **A9**: We chose SelectiveNet because of its stable performance without the need to tune additional hyperparameters. For example, as explained in Section 8.2 of [3], Deep Gamber’s performance highly depends on the value of the additional hyperparameter $o$: when $o$ is small, it could converge to a trivial solution which rejects all examples, especially on more difficult datasets. In addition, compared to other models, we have considered more difficult datasets such as CIFAR100.
>
> **Reference**.
> [1] Geifman, Yonatan et al. "Selectivenet: A deep neural network with an integrated reject option." ICML  2019.
> [2] Hacohen, Guy, et al. "On the power of curriculum learning in training deep networks." ICML  2019.
> [3] Liu, Ziyin, et al. "Deep gamblers: Learning to abstain with portfolio theory." NeurIPS 2019

---

### Official Review · Reviewer_QZ16 · 2022-10-26

**Confidence:** 4
**Correctness:** 3
**Technical Novelty And Significance:** 2
**Empirical Novelty And Significance:** 2
**Recommendation:** 3

**Clarity, Quality, Novelty And Reproducibility:**

### Novelty
The novelty comes from the idea that combines the selective classification problem and curriculum-inspired training. However, I evaluate it as a marginal novelty as it is not convincing enough why that combination is needed (as mentioned in Weakness - Rationale).

### Clarity
There were no big obstacles to understanding its main idea and experimental designs.

Minor)
1. There are missing references for the sentence "In many real life scenarios such as medical diagnosis, robotics and self driving cars ...".
2. If several tables (Table 1-4) are combined into one or two, the other interesting results can be moved from the supplementary to the main paper.

### Quality
The quality of the paper needs to be improved by providing enough experimental evidence to support the approach and to understand the model's behavior (as mentioned in Weakness - Hyperparameter choice and Ablation study).

### Reproducibility
This paper provides source codes for reproduction and uses publicly available datasets in their experiments.

**Strength And Weaknesses:**

- Rationale: It is not clear why we need to combine the two approaches (selective classification and curriculum-inspired training). The authors need to clearly state a rationale for their choice (i.e., the combination of the two). It can be an argument that the two methods are complement each other, or extensive experimental evidence showing that the combination of the two outperform each one.

- Ablation study: In addition to Inception (to provide a rationale for their difficulty score estimation) and Vanilla-Sekection (to provide a rationale for the mini-batch technique), another option can be added to provide a rationale for the selective classification technique in thier model: non-selective classification with the mini-batch technique.

- Hyperparameter choice: This approach uses several hyperparameters (e.g. coverage rates, alpha, lambda, etc.) that can impact the model performance. Therefore, several experiments are necessary to understand those. For example, all experimental results (Table1 - 4, Supp Table 6) show that the performance improves as it uses lower target coverage rates. Can you provide further experimental results to find what coverage rate is preferable for the model? Also, it would be good to see how the empirical coverage changes along with the target coverage in addition to the supplementary explanations.

- [Minor] Add missing references (see comment in Clarity - Minor 1)

- [Minor] Make the main paper more self-inclusive (see comment in Clarify Minor 2)




**Summary Of The Paper:**

This paper proposes a variation of SelectiveNet (Geifman & El-Yaniv, 2019) inspired by curriculum learning. The approach can be seen as a combination of selective classification and curriculum training. As illustrated in Figure 1, it aims to minimize a joint loss between the selective loss term constrained by the empirical coverage and standard loss. The authors provide several experimental results on benchmark datasets (CIFAR10, CIFAR100, SVHN) that their approach achieves lower error rates than the baseline (SelectiveNet). They also compare the performance with several variants of their methods to provide a rationale for their design choice.

**Summary Of The Review:**

Overall, this paper should be improved by
(1) providing a good rationale for their design choice (i.e., why do we need to combine the selective classification and curriculum training), and
(2) providing additional experiments to improve the quality and novelty of the paper.

---

> ### Author Response · Authors · 2022-11-19
> **author response**
>
> We thank the reviewer for the insightful comments. Below we address all the major comments, numbered as **C1** to **C5** (corresponding answers are numbered as **A1**-**A5**)
>
>
> **C1**: It is not clear why we need to combine the two approaches (selective classification and curriculum-inspired training). The authors need to clearly state a rationale for their choice (i.e., the combination of the two). It can be an argument that the two methods are complement each other, or extensive experimental evidence showing that the combination of the two outperform each one.
> **A1**: We apologize for not having made this clear enough. Briefly speaking, we combine these two approaches because of two benefits: (1) calibrating the ratio of easy and hard examples to mitigate the misguiding noise issue, (2) exploiting the difficulty ordering at the mini-batch level to improve the accuracy and convergence speed. We have explained these in the Contributions part of the Introduction, and further elaborated on them in Section 4.1 and 4.2. We believe that as long as there are tangible benefits, it’s worth studying how to combine them, which is what we are trying to achieve in our paper.
>
> **C2**: Ablation study: In addition to Inception (to provide a rationale for their difficulty score estimation) and Vanilla-Sekection (to provide a rationale for the mini-batch technique), another option can be added to provide a rationale for the selective classification technique in their model: non-selective classification with the mini-batch technique.
> **A2**: We study the problem of selective classification in our paper. Therefore, only those methods that are applicable to selective classification are considered in our experiments. As suggested by the reviewer, it’s certainly an interesting idea to investigate non-selective classification with the mini-batch technique. However, since it’s not designed for selective classification, we are afraid that we can’t run this method in our experimental settings.
>
> **C3**: Can you provide further experimental results to find what coverage rate is preferable for the model?
> **A3**: We thank the reviewer for bringing up this point. We would like to note that there is a trade-off between coverage and error rate: the larger the coverage rate is, the larger the error rate is. Our experimental results are consistent with this trend (see Table 1-4). Actually, this is an inherent characteristic for the selective classification problem [1]. Therefore, the preferable coverage rate depends on what error rate is desired. In other words, there is no single value for the preferable coverage rate, because different desired error rates lead to different coverage rates. Furthermore, according to Table 1-4, it’s easy to determine the preferable coverage rate for different error rates: scan the table to find the largest coverage rate without exceeding the desired error rate.
>
> **C4:** Also, it would be good to see how the empirical coverage changes along with the target coverage in addition to the supplementary explanations.
> **A4**: We would kindly note that Table 8 in our appendix exactly shows how the empirical coverage changes along with the target coverage. From Table 8, we can observe that the empirical coverage is highly correlated with the target coverage, and our method has smaller coverage violations.
>
> **C5**: There are missing references for the sentence "In many real life scenarios such as medical diagnosis, robotics and self-driving cars ...".
> **A5**: We are happy to add more references: “In many real life scenarios such as medical diagnosis [2], legal documents translation [3], robotics and self-driving cars [4]...”
>
> **Reference**.
> [1] Geifman, Yonatan, and Ran El-Yaniv. "Selective classification for deep neural networks." Advances in neural information processing systems 30 (2017).
> [2] Kompa, Benjamin, Jasper Snoek, and Andrew L. Beam. "Second opinion needed: communicating uncertainty in medical machine learning." NPJ Digital Medicine 4.1 (2021): 1-6.
> [3] Vieira, Lucas Nunes, Minako O’Hagan, and Carol O’Sullivan. "Understanding the societal impacts of machine translation: a critical review of the literature on medical and legal use cases." Information, Communication & Society 24.11 (2021): 1515-1532.
> [4] Ghodsi, Zahra, et al. "Generating and characterizing scenarios for safety testing of autonomous vehicles." 2021 IEEE Intelligent Vehicles Symposium (IV). IEEE, 2021.

---

### Public Comment · ~Qiang_Ding1 · 2022-11-08
**The idea of this paper is novel but the baseline is not strong**

The idea of this paper seems interesting. As far as I know, training a selective function is not easy. So it is reasonable to apply curriculum learning to the selective classifier in order to train the selective function more smoothly and easily.
However, the baseline SelectiveNet is not the state-of-the-art. The current state-of-the-art selective classifier is SAT [1]. I am curious about whether the curriculum-inspired routine proposed by this paper is effective on top of SAT?


[1] Lang Huang, Chao Zhang, and Hongyang Zhang. Self-adaptive training: beyond empirical risk minimization. Advances in Neural Information Processing Systems, 2020.

---

> ### Author Response · Authors · 2022-11-19
> **Thank you for the comment.**
>
> Thanks a lot for this comment. We are aware of other selective classifiers such as Deep Gamber and SAT, which are also discussed in our paper. However, we chose SelectiveNet because of its stable performance without the need to tune additional hyperparameters. Please also refer to our answer **A9** in our response to Reviewer xFiA.

---

### Decision · Program_Chairs · 2023-01-20

**Decision:**

Reject

**Justification For Why Not Higher Score:**

N/A

**Justification For Why Not Lower Score:**

N/A

**Metareview: Summary, Strengths And Weaknesses:**

The proposed approach in this paper combines elements of SelectiveNet and curriculum learning to address the challenge of selective classification. While the use of curriculum learning is a valuable contribution, the reliance on SelectiveNet as the only baseline is a limitation of the paper. Furthermore, the focus on Cifar and SVHN as the primary datasets is not as relevant or convincing as using the ImageNet dataset, which is widely considered to be more complex and challenging.

Overall, this paper has some valuable insights, but the limited empirical content and the lack of robust comparison to other baselines make it difficult to fully assess the effectiveness of the proposed approach. We encourage the authors to consider incorporating additional baselines and focusing on the ImageNet dataset in order to strengthen the empirical content of the paper and make it more convincing. Additionally, we recommend that the suggestions and questions raised by the reviewers should be carefully considered and addressed in a revised version of the paper.